# Testicular, Epididymal and Vasal Anomalies in Pediatric Patients with Cryptorchid Testes and Testes with Communicating Hydrocele

**DOI:** 10.3390/jcm11113015

**Published:** 2022-05-26

**Authors:** Jerzy Niedzielski, Maciej Nowak, Piotr Kucharski, Katarzyna Marchlewska, Jolanta Słowikowska-Hilczer

**Affiliations:** 1University Pediatric Centre, Department of Pediatric Surgery and Urology, Medical University of Lodz, 90-647 Lodz, Poland; piotr.kucharski@umed.lodz.pl; 2Post-Graduate Intern, Department of Pediatric Surgery and Urology, Medical University of Lodz, 90-647 Lodz, Poland; maciekrnowak@gmail.com; 3Department of Andrology and Reproductive Endocrinology, Medical University of Lodz, 90-647 Lodz, Poland; katarzyna.marchlewska@umed.lodz.pl (K.M.); jolanta.slowikowska-hilczer@umed.lodz.pl (J.S.-H.)

**Keywords:** undescended testis, intra-abdominal testis, canalicular testis, communicating hydrocele, testicular epididymal vasal anomalies

## Abstract

The goal of this study was to determine the prevalence of the testicular, epididymal, and vasal anomalies (TEVA) in cryptorchid and communicating hydrocele pediatric patients. Six hundred and ninety-one prepubertal boys underwent inguinal exploration for 741 undescended (UDT) or hydrocele testes. Two hundred and fifty-five TEVA were detected in 154 UDT boys, compared to 32 defects in 24 hydrocele patients (*p* < 0.001). The TEVA were more frequent in bilateral UDT (*p* = 0.009). Multiple defects were observed more frequently in the intra-abdominal testicles (*p* = 0.028). A correlation was found between the testicular atrophy index (TAI) and the incidence and number of TEVA in the UDT boys (*p* < 0.001). The smaller the testis (higher TAI), the more the defects that appeared in it and the higher the frequency of their appearance. Another correlation was established between testis position and the incidence and number of TEVA (*p* < 0.001). The higher the testis position, the more the defects that appeared in it and the higher the frequency of their appearance. A correlation was established between the position and the volume of the affected testis (*p* < 0.001). The higher the gonad position, the more severe the atrophy observed in it. The TEVA were more frequent in the UDT boys than in the hydrocele patients. We revealed that the risk of abnormal fusion between the testis, epididymis, and vas deferens is connected with the testis position (intra-abdominal testes) and bilateral non-descent.

## 1. Introduction

In human beings, as in most mammals, the testis descends from the abdomen to the scrotum (extracorporeal position) to find a lower ambient temperature for normal spermatogenesis [1]. The testicular descent occurs between the 8th and the 35th week of gestation in two stages, of which the trans-abdominal phase (8th–15th week of gestation) is rarely disrupted [2]. Only about 5% of undescended testes (UDTs) are intra-abdominal [3], while most of the descending testes are stopped at the inguinal canal during the inguino-scrotal phase [1].

The pathogenesis and etiology of cryptorchidism, although it has been extensively studied, is still unclear. Among multiple casual factors related to the condition, anatomical variations, including patent processus vaginalis (PV) and testicular/epididymal fusion anomalies, have been reported in cryptorchid patients, with a range from 32 to 72% [4,5]. However, it is not clear whether these anomalies cause non-descent of the testis or, conversely, whether they result from the cryptorchidism.

Cryptorchidism is frequently (66.7–73%) associated with patent PV [6,7]. The PV is a conduit extending from the peritoneum to the scrotum from around the 12th week of gestation which should be obliterated after the testicular migration is completed. If it does not close, the boy may develop an inguinal hernia or a communicating hydrocele [8,9].

Testicular and epididymal appendices have been considered to be congenital anomalies [8]. They are supposed to be the remnants of the cranial part of the Mȕllerian duct and the Wolffian duct [10]. Regression of the male Mȕllerian duct is mediated by the anti-mȕllerian hormone (AMH). Apoptosis of the Mȕllerian duct affects all its parts; thus, the origin of the testicular appendix can be the poor development of the fetal testis and its disturbed hormonal function. However, some studies report that these structures are present in most normal individuals [11]. It is suggested that they can control the amount of serous fluid in the vaginal tunica space; however, their function is still controversial [12].

The goal of this study was to determine the prevalence of the testicular, epididymal and vasal anomalies (TEVA) in cryptorchid patients compared with boys with a communicating hydrocele and to examine the correlation between the presence of TEVA and the position of the undescended testis.

## 2. Materials and Methods

### 2.1. Patients

We included 691 boys (aged 1.5 month–9 years) who were undergoing inguinal exploration for cryptorchidism and a communicating hydrocele in the Department of Pediatric Surgery and Urology at the University Center of Pediatrics in Lodz between January 2011 and December 2017. No patient was excluded from the study.

All the patients underwent clinical and ultrasound examination before the treatment. The position and the three dimensions of the testis were recorded and used to calculate the testicular volume (TV) and testicular atrophy index (TAI) of the affected testis [1]. The TAI of the bilaterally undescended testes was calculated in relation to the median TV of healthy gonads in the age group. Testes with TAI > 30% were considered as hypotrophic. All the patients were otherwise healthy, and all were prepubertal (Tanner I).

The patients were divided into age subgroups based on the developmental changes in testes: 0–2 years, 3–6 years, and 7–9 years [13] (Table 1). Another division was based on the types of disorder: intra-abdominal and canalicular UDT and communicating hydrocele.

### 2.2. Surgical Procedures

A total of 741 testes were operated on. During the surgery, via an inguinal approach, the spermatic cord was isolated, the hernial sac or PV was dissected, when present, and high ligation was performed. In cryptorchid patients, the testis’ location and size were examined; orchidofunicolysis and orchidopexy were performed by the conventional technique of placing the testis in the subdartos pouch [1]. With the communicating hydrocele, the distal part of the PV was cut open all the way down to the tunica testis and left open after evacuation of the fluid. As an initial procedure, boys with impalpable testes underwent diagnostic laparoscopy followed by a Fowler–Stephens operation (intra-abdominal testes) [1].

Patients with hydrocele were qualified for operation after twelve months of age. Early surgery was performed in boys with a concomitant inguinal hernia or a large persistent hydrocele hindering the daily care of a child [14].

The testis, epididymis, and vas deferens were meticulously examined, and any detected anomalies were recorded and classified as major (occluded sperm pathways—defects that could impair patient’s future fertility) or minor (patent sperm pathways—not influencing future fertility) (Table 2).

The anatomy of the gonads was regarded as normal when a normal firm attachment between the testis and the caput and cauda epididymis, as well as a connection to the vas deferens, was present. The anomalies were classified as a separation of the caput and/or cauda epididymis from the testis (Figure 1a,b) or a long looping epididymis/vas deferens (Figure 2) or other with a possible disruption (loss) of the testis-epididymis-vas connection (continuity) [15,16]. Testicular and epididymal appendices have not been counted as congenital anomalies [8,11] and were classified separately (Table 2).

### 2.3. Statistics

All the analyses were performed using Statistica 13.1 for Windows (StatSoft Inc., Tulsa, OK, USA). The distribution of the data was analyzed using the Shapiro–Wilk test. The data were distributed in a nonparametric manner; so, they were presented as median and range and analyzed using the ANOVA Kruskall–Wallis test for the assessment of the statistical difference between the groups. The Spearman r correlation was conducted between the results of the examined parameters. The Student *t*-test and Pearson’s chi-squared test were used for comparison of the examined parameter values and the incidence in the studied groups of patients. Differences were considered significant at *p* < 0.05.

## 3. Results

A total of 255 anomalies of the testis, epididymis, and vas deferens were detected in 164 UDT boys, compared to 32 defects in 24 patients with hydrocele (Table 2).

The overall incidence of TEVA (excluding appendix testis) was significantly higher in both UDT groups in comparison to the hydrocele group (100% and 33.4% vs. 9.6%, *p* < 0.001). Conversely, the incidence of appendices testis/epididymis was significantly higher in hydrocele patients (40.6% vs. 7.0 and 18.3%, *p* < 0.001, *p* = 0.002) (Table 2).

The incidence of major defects decreased with the lower position of the gonad, being significantly the highest in the patients with intra-abdominal testes and the lowest in the hydrocele group with testes in the scrotal position (81.4% vs. 16.5% vs. 2%, *p* < 0.001). By analogy, the incidence of the minor defects also decreased with the lower position of the testis (88.4% vs. 16.9% vs. 7.6%, *p* < 0.001, *p* = 0.004) (Table 2). Conversely, the incidence of appendices testis/epididymis increased with the lower position of the testis.

In 133 UDT patients, there was 1 defect only; in 17 patients, there were 2 defects; and in 14 patients, 3 defects were detected. One anomaly was observed in 16 boys, and 2 were observed in 8 boys with hydrocele (Table 3). Testes with only one TEVA predominated in the canalicular UDT group (94%, *p* < 0.001) and the hydrocele group (66.7%, *p* = 0.001), while in the intra-abdominal UDT group testicles with multiple defects were observed more frequently (72.1%, *p* = 0.028).

A significant correlation was found between the size of the testicle determined by the TAI and the incidence and number of TEVA, both in intra-abdominal (R = 0.9, *p* < 0.001) and canalicular (R = 0.55, *p* < 0.001) groups, as well as in the overall UDT group (R = 0.65, *p* < 0.001) (Figure 3.). The smaller the gonad (higher TAI), the more the defects that appeared in it and the higher the frequency of their appearance.

A significant correlation was also established also between the position of the testicle and the incidence and the number of TEVA (R = 0.5, *p* < 0.001). The higher the gonad position, the more the defects that appeared in it and the higher the frequency of their appearance.

In the intra-abdominal UDT group, all unilateral and bilateral gonads were affected (Table 4). However, in the canalicular UDT, as well as in the entire UDT group, TEVA were observed significantly more frequently in the bilateral undescended testes (48.8%, *p* = 0.036; 58%, *p* = 0.009).

The TAI of the undescended testes was significantly higher for the intra-abdominal gonads (*p* < 0.001) (Table 5). A significant correlation was established between the position and volume of the affected testis (TV) (R = 0.4, *p* < 0.001). The higher the gonad position, the more severe the atrophy that was observed in it.

Direct comparisons of the healthy and undescended testes’ size between the groups were not performed as these would have been directly dependent upon the patients’ age. The TAI seems to be the best tool for comparison, indicating the percentage loss of the affected testicle volume in relation to the healthy testicle, regardless of the absolute volume of the testicle related to the patient’s age.

## 4. Discussion

The sexual differentiation of male internal sex organs occurs in a narrow window of time between 8 and 12 weeks of human fetal development [18]. Leydig cells form outside the seminiferous tubules of the fetal testis and produce testosterone, which is secreted in an exocrine manner down the Wolffian duct, stimulating it to persist and form the epididymis, vas deferens, and seminal vesicles. To ensure that the testis, epididymis, and vas deferens will establish proper connections between themselves to create the future way of the semen transport, the epididymis head must be adjacent to the upper pole of the testis, the epididymal duct must be connected to the vas deferens, and the vas deferens itself must run continuously up to its opening in the prostatic urethra [19]. In order for the testis, epididymis, and vas deferens to serve the function of producing, maturing, and transporting the male germ cells, they must undergo a highly coordinated succession of molecular and morphogenic events during development. Disturbances may appear at any moment of these transformations, which will result in defects in the organs and the connections between them. These in turn may be either irrelevant to the function of the entire system or may impair or even prevent the production and transport of semen in the adult man [1,20]. Moreover, fetal Leydig cells produce insulin-like hormone 3 (INSL3), which stimulates the growth of the genito-inguinal ligament (gubernaculum), which is important for the first phase of testicular descent [2].

Our results are consistent with those of other authors who have shown that cryptorchidism can be associated with various anatomical anomalies, but epididymal anomalies and patency of the PV are among the most frequent [4,5,21]. Rachmani et al. [22] observed that only complete nonfusion of the epididymis seems to reliably interfere with epididymal-testicular descent. We did not find such a relationship.

Moreover, we revealed that the higher the position of the testis the greater the incidence of TEVA. The same observation was made by Caterino et al. [17], who found out that epididymal/testicular fusion anomalies and the persistence of a patent PV were strongly associated with a more proximal position of the testis. Testes located in the higher position frequently have a lower volume and histological signs of dysgenesis; in addition, men with such a condition have problems with fertility [23,24,25]. These findings may suggest that TEVA may also be the result of poor development of the testes and their disturbed function during the fetal period of life. It has been shown that the PV at birth is patent in 80% of children and progressively closes during the first year of life under the influence of androgens [26].

Thus, we may assume that some factors secreted by developing testes (predominantly testosterone) are required not only for the epididymal development, but also for proper connections with the testicular efferent ducts, as well as PV obliteration. The expression of androgen and estrogen receptors in the epithelia of the early efferent ducts has been well documented [20]. However, Kraft et al. [27] have shown that fusion anomalies, although associated with smaller testes, are not associated with significant abnormalities in the germ cell number and testicular histological structure. In turn, Anderson et al. [28] observed a significantly higher prevalence of epididymal anomalies (14%) and patent PV (21%) even in retractile testes. Over 70% of patients with this condition show normal testicular development and function; however, in some cases morphological alterations in the testicular structure were found [29], as well as poor semen parameters in adulthood [30]. The implication is that the pathogenesis of TEVA, in a similar way to the pathogenesis of cryptorchidism, may be the result of testicular dysgenesis, but not in all cases however. However, Han et al. [29] had a different observation, which was that the patency of PV was strongly associated with the epididymal anomalies irrespective of testicular descent.

In our study, we observed that boys with hydrocele show, although less frequently, TEVA as an accompanying disorder. This was observed by other authors as well [4,5]. In turn, in hydrocele patients we noted the significantly higher incidence of testis/epididymis appendices in comparison to the UDT patients. This is in line with studies which have hypothesized about the possible role of the appendix testis in testicular descent, noticing a reduced incidence of appendix testis among boys with UDT compared to boys without testicular maldescent [31,32]. The testis appendix was considered as an organ taking part in fluid regulation within the vaginal tunica space [33]. The higher incidence of testis/epididymis appendices in patients with hydrocele may suggest a defect in fluid regulation, especially in those patients with accompanying TEVA (9.6%). However, such a hypothesis needs further elucidation.

## 5. Conclusions

In our study, we confirmed that TEVA are frequent in boys with UDT, more than in boys with hydrocele. We revealed that the risk of abnormal fusion between the testis, epididymis, and vas deferens correlates with the position and hypotrophy and with the bilateral non-descent of testis and is the highest in intra-abdominal, hypotrophic, and bilateral undescended testes. These abnormalities may be related to underlying testicular dysgenesis and androgen deficiency in utero rather than to the patent PV. However, this observation is not the explanation for the pathogenesis of UDT and TEVA in all cases. As major TEVA may impair future fertility, patients with a history of cryptorchidism should be under the care of an andrologist after the age of 18.

## Figures and Tables

**Figure 1 jcm-11-03015-f001:**
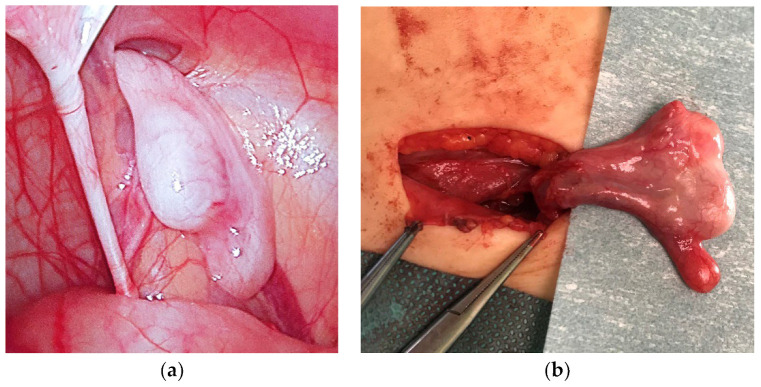
A boy with unpalpable right testis: (**a**) diagnostic laparoscopy at 9 months of age—an intra-abdominal testis with a separation of caput epididymis is visible; (**b**) the same testis during canalicular orchiopexy at 16 months of age (second stage of Fowler–Stephens operation).

**Figure 2 jcm-11-03015-f002:**
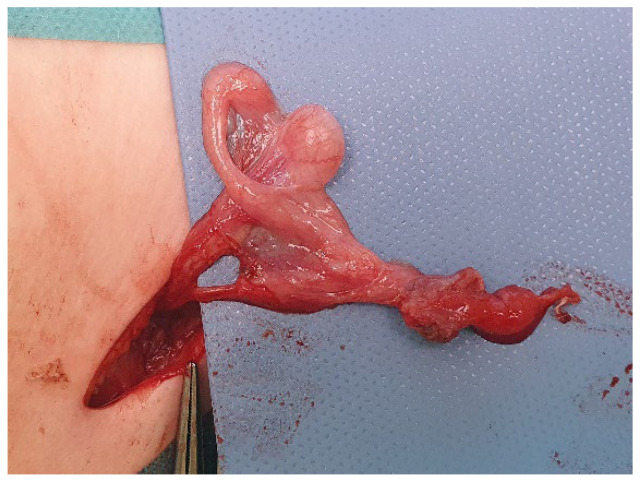
A boy with right canalicular undescended testis: orchiopexy at 13 months of age—a testis with the complete separation of epididymis (caput and cauda) and “long looping” vas deferens is visible.

**Figure 3 jcm-11-03015-f003:**
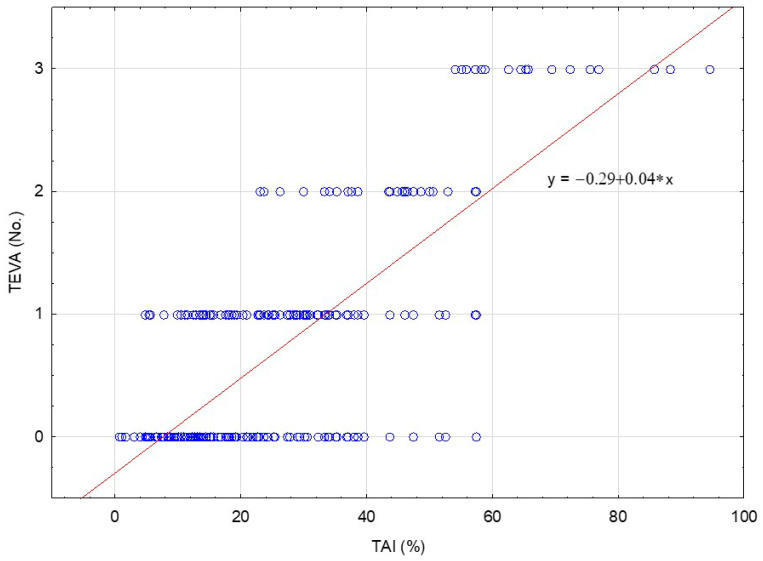
Spearman’s rank correlation between TAI and number of TEVA in the entire group of UDT patients (R = 0.6490, *p* < 0.001). Red line indicates trend of linearly related variables.

**Table 1 jcm-11-03015-t001:** Patients’ characteristics.

	Intra-Abdominal UDT	Canalicular UDT	Communicating Hydrocele
Age (years)	0.1–2.0	0.9–9.0	1.0–5.0
Mean ± SD	0.7 ± 0.5	4.2 ± 2.6	2.7 ± 1.3
Median	0.6	4.0	3.0
Age groups (n)			
0–2 years	34 (100%)	144 (35.3%)	118 (47.4%)
3–6 years	-	166 (40.7%)	131 (52.6%)
7–9 years	-	98 (24.0%)	-
Total (n)	34 (100%)	408 (100%)	249 (100%)
Testes (n)	43	449	249
Side (n)			
R	12 (35.3%)	209 (51.2%)	118 (47.4%)
L	13 (38.2%)	158 (38.7%)	131 (52.6%)
Bil	9 (26.5%)	41 (10.1%)	-

UDT—undescended testis, n—number of cases, R—right, L—left, Bil—bilateral.

**Table 2 jcm-11-03015-t002:** Classification and distribution of testicular, epididymal, and vasal anomalies—TEVA [15,16,17].

	1. Intra-Abdominal UDT (n = 34; t = 43) a/n/t (%)	2. Canalicular UDT (n = 408; t = 449) a/n/t (%)	3. Communicating Hydrocele (n = 249; t = 249) a/n/t (%)	*p*
**Major (anomalies/boys)**	**57/28/35 (81.4)**	**78/66/74 (16.5)**	**5/5 (2)**	***p* < 0.001**
A. testicular hypotrophy/hypoplasia incl. atrophy/agenesis (TAI > 30%)	35 (81.4)	44 (9.8%)	3 (1%)	*p* < 0.001
B. epididymal atrophy	1	3	0	
C. separation caput epididymis, complete separation	20	18	2	
D. lack of continuity or hypotrophy of vas deferens	1	13	0	
**Minor (anomalies/boys)**	**38/29/38 (88.4)**	**82 /64/76 (16.9)**	**27/19 (7.6)**	**1–2: *p* < 0.001**
A. separation cauda epididymis	38	62	19	**2–3: *p* = 0.004** **1–3: *p* < 0.001**
B. epididymal cysts	0	11	6	
C. long looping epididymis/ vas deferens	0	9	2	
**Total**	**95/34/43 (100)**	**160/130/150 (33.4)**	**32/24 (9.6)**	***p* < 0.001**
**Appendix testis/epididymis**	**3/3/3 (7.0)**	**82/70/82 (18.3)**	**101/101 (40.6)**	**1–2: NS** **2–3: *p* < 0.001** **1–3: *p* = 0.002**

NS—not significant, TAI—testicular atrophy index, UDT—undescended testis, a—number of anomalies, n—number of patients, t—number of testes, (%)—incidence of TEVA was calculated by number of affected testes to total number of testes; Pearson’s chi-squared test.

**Table 3 jcm-11-03015-t003:** Distribution and number of TEVA per one gonad and one patient.

No of TEVA in One Patient/Testis	Intra-Abdominal UDT n/t (%)	Canalicular UDT n/t (%)	Communicating Hydrocele n/t (%)
One	11/12 (27.9)	122/141 (94)	16/16 (66.7)
Two	10/10 (23.3)	7/8 (5.3)	8/8 (33.3)
Three	13/21 (48.8)	1/1 (0.7)	0/0
Total	34/43 (100)	130/150 (100)	24/24 (100)

n—number of affected patients, t—number of affected testes, (%)—distribution of TEVA was calculated by number of testes with one or more anomalies to total number of affected testes.

**Table 4 jcm-11-03015-t004:** Gonads with TEVA regarding unilateral or bilateral UDT.

Type of UDT	Intra-Abdominal t/tt (%)	Canalicular t/tt (%)	Total t/tt (%)
Unilateral	25/25 (100)	110/367 (30)	135/392 (34.4)
Bilateral	18/18 (100)	40/82 (48.8)	58/100 (58)
Total	43/43 (100)	150/449 (33.4)	193/492 (39.2)

UDT—undescended testis, t—number of affected testes, tt—total numer of testes, (%)—distribution of TEVA was calculated by number of affected testes to total number of unilateral or bilateral UDT.

**Table 5 jcm-11-03015-t005:** TV and TAI of undescended gonads in intra-abdominal and canalicular UDT patients.

Undescended Testis	Intra-Abdominal UDT	Canalicular UDT	*p*
TV median (±SD)	0.18 (±0.07)	0.41 (±0.29)	
min–max	0.02–0.37	0.11–1.57	
TAI median (±SD)	**54.08 (±18.69)**	**16.73 (±12.28)**	***p* < 0.001**
min–max	18.18–94.51	0.53–57.45	

TV—testicular volume, TAI—testicular atrophy index; Pearson’s chi-squared test.

## Data Availability

The data presented in this study are available on request from the corresponding author. The data are not publicly available until primary analyses for the other outcomes of this study are completed.

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
