# Peer review of "Testicular, Epididymal and Vasal Anomalies in Pediatric Patients with Cryptorchid Testes and Testes with Communicating Hydrocele"

_jcm, 2022, doi:10.3390/jcm11113015_

Round 1
Reviewer 1 Report
In this manuscript the authors sought out to evaluate the prevalence of the testicular, epididymal and vasal anomalies (TEVA) in in cryptorchid patients compared with boys with communicating hydrocele. 691 patients were included.
The analysis seems solid and appropriate.
A significant correlation was found between the testicular atrophy index and the incidence and number of TEVA in undescended testes patients. Similarly, a strong correlation was found between testis position and the incidence and number of TEVA: the higher the testis position, the more frequently and more defects appeared in it. Overall interesting resuts potentially informing future surgical treatment and follow-up in patients who harbored such risk factors and syndromes.
Incidental testicular lesions can occur in the follow-up of these patients and organ-sparing approaches have been advocated (Gentile et al. doi: 10.1097/JU.0000000000000579; Narayan et al. doi: 10.4081/aiua.2021.3.296; Ory et al. doi: 10.1002/bco2.77). Could the authors further discuss the implications of these results in the follow-up of such patients considering this particular setting? Tailored follow-up proposal?
Author Response
Thank you very much for this suggestions, however incidental testicular lesions were not the aim of our study.
Reviewer 2 Report
What is the main question addressed by the research?
This study describes Testicular, Epididymal and Vasal Anomalies (TEVA) observed in 691 prepubertal boys (with a wide range of age: 1.5 month - 9 years old) who underwent surgical exploration for undescended testes or communicating hydrocele.
Is it relevant and interesting?
They found interesting data. The size of the testis (measured as Testicular index atrophy) and testis position were correlated in reverse: the higher was gonad position, the more severe atrophy was observed. Furthermore, each parameter was correlated with incidence and number of TEVA. Also, TEVA were more frequent in undescended testes than in hydrocele patients. In my opinion this research it is useful to understand occurence of TEVA in two different clinical scenarios, that are frequently observed in clinical practice.
How original is the topic?
This is an observational study on anatomical anomalies, therefore originalty is limited. They added a careful evaluation in each case.
What does it add to the subject area compared with other published material ?
I think yes, but I am dedicated to adult uro-adrology, therefore my competence in these field of literature is quite limited.
Is the paper well written?
Yes, however it needs english language editing.
Is the text clear and easy to read?
Its clear, but it is not easy to read. This data are well supported and clearly described. Although several groups of age and multiple correlations were made, an excessive analysis in subgroups is not easy to understand and read. Authors reported all possible correlations, that lead to a long manuscript loosing sight of main observations.
Are the conclusions consistent with the evidence and arguments presented? Yes, The population observed and analysis done were enough to drawn and support the conclusions
Do they address the main question posed?
Yes. Main message are well summarized and supported.
I hope to have addressed adequately my opinion and comments
Author Response
Thank you for the opinion.
Our analysis is indeed detailed and multi-parameter. Our intention was to show and investigate which factors (location of the testicle, side, age of the patient, etc.) have and which do not affect the presence of the examined defects. The more so, because we have not found a similar multidirectional analysis in the available literature. We consider this as an advantage rather than a disadvantage of our work.
Reviewer 3 Report
Dear colleague,
Your work comes to draw attention upon a current subject yet incompletely understood: the undescended testis and the associated testicular, epididymal and vascular anomalies.
In your article you studied the prevalence of these anomalies in a significant group of patients and made a comparison to another group of children who needed surgical treatment for other pathology of the inguinal canal. You recorded and classified as major or minor every testicular, vascular and/or epididymal anomaly and found a correlation between the presence of these anomalies and the position of the undescended testis.
Your results increase the current available information and your observations will give opportunities for future studies to establish further connections and understanding the mechanisms which make a testicle stop its descent, a processus vaginalis to stay patent and a male with undescended testis to have fertility problems or not.
After carefully reading your work I find useful to mention a few comments on the article to make it in its best version:
- For the information to be more easily read, make wider columns in Table 1, 3, 4 and 5.
- For a better and quick follow of the results, put the abbreviation index below the table (and not on the next page) in Table 2.
- Mention if there were special criteria of including/excluding patients in the study (associated pathology…)
- For the control group- what were the indications for the surgical treatment of the hydrocele?
- In the discussion chapter give more details about the impact of finding testicular, epididymal and vascular anomalies during surgical treatment and their applications in clinical practice (fertility etc.)
- Are there any significant differences between age groups to be mentioned and discussed separately (age-related differences regarding the severity and number of anomalies)?
- What do you consider as being the limits of your study ?
- What further research directions does your study open?
- You should develop a bit more your conclusions referring your results to current literature.
Author Response
- For the information to be more easily read, make wider columns in Table 1, 3, 4 and 5.
Thanks for your suggestions. We think that such changes will be introduced by the editor of the journal if our article is qualified for publication.
- For a better and quick follow of the results, put the abbreviation index below the table (and not on the next page) in Table 2.
The suggested changes were made.
- Mention if there were special criteria of including/excluding patients in the study (associated pathology…).
There were no specific criteria for including or excluding patients from the study groups.
The note has been placed in the chapter Material and Methods.
- For the control group- what were the indications for the surgical treatment of the hydrocele?
According to EAU Guidelines on Paediatric Urology 2022, persistence of a simple scrotal
hydrocele beyond twelve months of age is an indication for surgical correction. Early surgery is indicated if there is suspicion of a concomitant inguinal hernia or underlying testicular pathology or a large persistent hydrocele hindering the daily care of a child.
The note has been placed in the chapter Material and Methods.
- In the discussion chapter give more details about the impact of finding testicular, epididymal and vascular anomalies during surgical treatment and their applications in clinical practice (fertility etc.).
The note has been placed in the chapter Conclusions.
- Are there any significant differences between age groups to be mentioned and discussed separately (age-related differences regarding the severity and number of anomalies)?
Our analysis showed that there are no differences between the age subgroups of patients in the incidence and distribution of TEVA. Which was to be expected, since TEVA are formed in the period of organogenesis and the postnatal period of gonad development no longer affects their frequency and distribution.
- What do you consider as being the limits of your study ?
We have not found and encountered any significant limitation in our research and analysis performed.
- What further research directions does your study open?
The natural direction of continuing our research seems to be to clarify the cause-and-effect relationship between the non-descent of the testis and the presence of TEVA. Which of them is a consequence and which is a cause? As we mentioned in our work, so far there is no idea of a methodology for testing the above relationship.
- You should develop a bit more your conclusions referring your results to current literature.
Our results are discussed in relation to the current literature in the chapter Discussion.